# Experimental and Theoretical Studies of the Optical Properties of the Schiff Bases and Their Materials Obtained from o-Phenylenediamine

**DOI:** 10.3390/molecules27217396

**Published:** 2022-10-31

**Authors:** Magdalena Barwiolek, Dominika Jankowska, Anna Kaczmarek-Kędziera, Slawomir Wojtulewski, Lukasz Skowroński, Tomasz Rerek, Paweł Popielarski, Tadeusz M. Muziol

**Affiliations:** 1Faculty of Chemistry, Nicolaus Copernicus University in Torun, Gagarina 7, 87-100 Torun, Poland; 2Faculty of Chemistry, University of Białystok, Ciolkowskiego 1K, 15-245 Białystok, Poland; 3Faculty of Chemical Technology and Engineering, Bydgoszcz University of Science and Technology, Kaliskiego 7, 85-796 Bydgoszcz, Poland; 4Institute of Physics, Kazimierz Wielki University, Chodkiewicza 30, 85-064 Bydgoszcz, Poland

**Keywords:** Schiff bases, ellipsometry, DFT, fluorescence, X-ray, thin layer, Hirshfeld analysis

## Abstract

Two macrocyclic Schiff bases derived from o-phenylenediamine and 2-hydroxy-5-methylisophthalaldehyde **L1** or 2-hydroxy-5-tert-butyl-1,3-benzenedicarboxaldehyde **L2**, respectively, were obtained and characterized by X-ray crystallography and spectroscopy (UV-vis, fluorescence and IR). X-ray crystal structure determination and DFT calculations for compounds confirmed their geometry in solution and in the solid phase. Moreover, intermolecular interactions in the crystal structure of **L1** and **L2** were analyzed using 3D Hirshfeld surfaces and the related 2D fingerprint plots. The 3D Hirschfeld analyses show that the most numerous interactions were found between hydrogen atoms. A considerable number of such interactions are justified by the presence of bulk *tert*-butyl groups in **L2**. The luminescence of **L1** and **L2** in various solvents and in the solid state was studied. In general, the quantum efficiency between 0.14 and 0.70 was noted. The increase in the quantum efficiency with the solvent polarity in the case of **L1** was observed (λ_ex_ = 350 nm). For **L2**, this trend is similar, except for the chloroform. In the solid state, emission was registered at 552 nm and 561 nm (λ_ex_ = 350 nm) for **L1** and **L2**, respectively. Thin layers of the studied compounds were deposited on Si(111) by the spin coating method or by thermal vapor deposition and studied by scanning electron microscopy (SEM/EDS), atomic force microscopy (AFM), spectroscopic ellipsometry and fluorescence spectroscopy. The ellipsometric analysis of thin materials obtained by thermal vapor deposition showed that the band-gap energy was 3.45 ± 0.02 eV (359 ± 2 nm) and 3.29 ± 0.02 eV (377 ± 2 nm) for **L1**/Si and **L2**/Si samples, respectively. Furthermore, the materials of the **L1**/Si and **L2**/Si exhibited broad emission. This feature can allow for using these compounds in LED diodes.

## 1. Introduction

Schiff base ligands reveal a broad variety of coordination architectures and structural differentiation. The presence and position of the azomethine group in macrocyclic compounds allow the formation of multi-donor ligands. Macrocyclization processes are ensured by the diversity of the structures of the resulting ligand. Thus, they give stable metal complexes with many metal ions. Macrocyclic Schiff bases are used in the fields of biochemistry [1,2,3], supramolecular chemistry [4,5], materials science [6], molecular recognition [7,8], and catalysis [9,10].

The multi-donor macrocyclic Schiff bases (e.g., S, N, O) can form compounds of various structures, which are crucial in coordination chemistry, as they can influence the structural, magnetic, or optical properties of the obtained complexes [11,12,13,14,15,16,17]. The isolation of mono-, bi-, and polynuclear complexes is possible [18,19]. Moreover, it is well known that Schiff base ligands containing S, N, O donor groups and chromophores, such as the azomethine group, are the basis for the formation of the high luminescence compounds [20,21,22,23,24,25,26]. The multi-donor macrocyclic Schiff bases combine the chemical, electronic and optical properties with those of the organic materials. So, even the subtle changes in the electronic or structural properties of the Schiff bases can result in obtaining a new group of interesting compounds, which can be used as new functional materials with applicable mechanical, thermal, chemical and optoelectronic properties [11,14. In inorganic as well as coordination chemistry, a considerable part of compounds not only have a wide variety of possible donor atoms but can also tune their properties through possible modification of their molecular structures. The introduction of various substituents or rings into the macrocyclic skeleton allows for the modification of the structural and spectroscopic properties of the isolated new compounds [27,28,29,30].

It has been intimately researched that Schiff base ligands are also organic sensors with selectivity for the different metal ions; they detect both anions and cations [31,32,33,34]. Two zinc(II) complexes obtained from differently O-substituted imidazole based homologous Schiff bases: 2-((E)-(3-(1H-imidazole-1-yl)propylimino)methyl)-6-ethoxyphenol) and 2-((E)-(3-(1H-imidazole-1-yl)propylimino)methyl)-6-methoxyphenol) were synthesized. Their sensitivity and selectivity toward arsenate were checked [33]. Interestingly, despite the small structural differences, the various sensitive fluorescence behavior toward arsenic was noted. It was a consequence of the presence of ethoxy and methoxy groups in the Schiff base rings; hence, due to the steric crowding in 2-((E)-(3-(1H-imidazole-1-yl)propylimino)methyl)-6-ethoxyphenol) no fluorescence changing were noted.

Additionally, Schiff bases could create thin fluorescent films using the spin, dip coating, Langmuir-Blodgett or organic vapor methods [35]. Films can provide thin design and high luminescence and improve numerous parameters devices such as OLED and photovoltaic [36]. Fluorescent organic materials are attracting interest in optoelectronics and cellular imaging. Moreover, organic materials give emission at a specific wavelength, and certain materials exhibit solvatochromism, which is the change in the optical properties of the material upon a change of the solvent polarity. The choice of a suitable compound for designing new materials is one of the most important challenges in the synthesis and application of new units with different and unusual properties are facing. There are still many questions regarding the fluorescence properties of compounds and thin films. The appropriate methods of establishing new materials should also be developed. New films can improve key parameters such as high luminescence, thin designs, and provide new unique characteristics of the new devices such as smartphones, OLEDS or solar batteries.

In the present work, we report the synthesis of the two macrocyclic ligands obtained in the reaction of o-phenylenediamine and 2-hydroxy-5-methylisophthalaldehyde **L1** or 2-hydroxy-5-tertbutyl-1,3-benzenedicarboxaldehyde **L2** (Figure 1).

DFT calculations were carried out to support the interpretation of the results concerning the optical properties of the studied ligands. Hirshfeld analysis extensively shows intramolecular interactions like it was observed for other compounds [37,38]. The emissive ligands cover by the spin coating and thermal vapor methods create thin fluorescent films. The morphology of the layers was analyzed by AFM and SEM microscopy.

## 2. Results and Discussion

### 2.1. Ligand Synthesis and Characterization

The 1:1 reaction of the corresponding aldehyde and primary amine forms the **L1** and **L2** ligands in yields ranging between 50% and 60% (Figure 1). The ligands were characterized by ^1^H, ^13^C, ^1^H ^13^C hmqc and hmbc NMR, UV-vis, IR and the elemental analysis. ^1^H, ^13^C NMR spectra are presented in Appendix A. On the ^1^H NMR spectra of both ligands, signals from azomethine hydrogen bonds between 8.55 and 8.65 ppm appeared. Moreover, signals from OH at 13.50 pm and 13.57 ppm for **L1** and **L2**, respectively, were noted. Resonances from aromatic rings were observed in the range of 6.31–7.18 ppm for **L1** and 6.81–7.45 ppm for **L2**, as expected. Additionally, at ^1^H NMR spectra, signals from -NH-CH_2_-(6) (Appendix A) at 4.40 ppm for **L1** and from -NH-CH_2_-(7) at 4.48 ppm (Appendix A) for **L2** were registered. Moreover, at ^13^C NMR spectrum, signals are observed at 46.91 ppm (C6) for **L1** as well as at 47.37 ppm (C7) for **L2** (Appendix A, respectively). The same was observed previously [27,28]. The presence of the signals of both -N=CH- and -NH-CH_2_- groups confirmed obtaining partially reduced macrocycles in which two C=N groups have been reduced to -CH_2_-NH ones.

The ligands are stable in air and soluble in several solvents, such as chloroform, methanol, acetonitrile, and benzene. The IR spectra displayed in Appendix A exhibit peaks at 1600 and 1587 cm^−1^ from stretching vibrations of the azomethine group. A sharp band at 3405 cm^−1^ for **L1** and 3423 cm^−1^ for **L2**, can be ascribed to the v_NH_ groups [11,29]. The peaks at 3405, 3423 cm^−1^ also are attributed to the OH group vibrations [29]. Additionally, bands from Ph-O stretching vibrations in the range of 1289–1288 cm^−1^ were noted. The elemental analysis confirmed the purity of the obtained compounds.

### 2.2. Crystal Structure Description—Hirshfeld Analysis

#### 2.2.1. **L1**

In packing along the b axis, we observe columns composed solely of O14 or O34 molecules (Figure 2 and Appendix A). Motifs from adjacent columns form a zipper of π-π interactions between rings coming from two different molecules. There were found only intramolecular N-H…O and O-H…N hydrogen bonding. Hence, tightly packed molecules form mainly weak interactions (packing index 69.3%) [39]. The most numerous interactions were found between hydrogen atoms, whereas the shortest contacts were created between hydrogen and carbon atoms (Figure 3 and Appendix A). In the case of O14 molecule, the red spots on the Hirshfeld surface are related to C…H interactions, whereas for O34 molecule they come from the complementary H…C interactions proving that mainly O14…O34 interactions are observed.

#### 2.2.2. **L2**

In packing along c axis, we observe a column composed of alternately arranged O14 and O34 molecules rotated by 46° (Figure 4 and Figure S9). Stacking interactions between aromatic rings coming from two different superposed molecules were detected, whereas N-H…N and O-H…N hydrogen bonds are solely intramolecular. Hence, in the crystal network, tightly packed molecules (packing index 69.8%) form mainly weak interactions (Figure 4).

The most numerous interactions were found between hydrogen atoms, whereas the shortest contacts were created between hydrogen and carbon atoms. In the former case, a huge number of such interactions (even greater than for **L1**) is justified by the presence of bulk *tert*-butyl groups. H…H contacts occur as spikes on the fingerprints (Figure 5 and Appendix A). These groups assure separation between adjacent molecules in the column as well as are involved in numerous weak contacts. In the latter case, red spots on the Hirshfeld surface are related to wings observed on the corresponding fingerprints. It should be noted that significantly different fingerprints for O14 and O34 molecules prove that both moieties form different interaction patterns.

### 2.3. UV-Vis and Fluorescence Spectroscopy

The absorption of the UV-Vis spectra and the fluorescence of the ligands were recorded at room temperature in various solvents showing polar and non-polar properties, successively MeCN, chloroform, methanol and benzene. (Figure 6, Appendix A).

The UV-Vis spectra of the **L1** and **L2** compounds showed bands in the range of 336–350 nm for **L1** and 338–346 nm for **L2**, from the π→π* transitions of the azomethine group (Figure 6, Appendix A), which was confirmed by DFT data (see below). In the **L1** and **L2** spectra between 382 and 390 nm in all solvents except methanol, bands related to the intra-ligand transitions appeared. The bands are attributed based on extinction coefficients.

The absorption UV-Vis spectra of the ligands were also recorded in the solid state at room temperature. (Figure 7). At the spectra, the one broad band divided into three components can be noted. They are connected with π→π*, IL charge transfer transitions in the azomethine, -NH-CH_2_- bonds and aromatic rings. In comparison with the solutions spectra, the bathochromic shift of the absorption bands (250, 344, 482, 506 nm) in the solid state like it was previously observed by us [30]. It is connected with the highest rigidity of the molecular skeleton of the cyclic molecule.

The interpretation of the experimental spectra of both Schiff bases was additionally confirmed by the DFT calculations (Appendix A, Appendix A). Theoretical results indicate that the absorption spectra for **L1** and **L2** are comparable to each other due to a similar structure of the molecular skeleton of both molecules and the nature of the consequent excitations. **L1** exhibits the longest wavelength signal at 420 nm. This transition corresponds to the HOMO→LUMO excitation of the π→π* character (compare the orbital shapes in Appendix A). In the shorter wavelength range, additional multiple transitions are present, contributing to the intensive band below 320 nm (Appendix A). For **L2**, the most intensive band is hypsochromically shifted with respect to the **L1** spectrum and appears at 403 nm. The frontier molecular orbitals for **L2** are presented in Appendix A.

The studies of the fluorescence properties of the compounds showed that the excitation of **L1** at 295 nm leads to the emission in the range of 454–516 nm, and of **L2** to the emission between 452 and 517 nm in dependence on the solvent polarity. However, when excitation was set at λ_em_ = 350 nm, the stronger intensity of the emission bands was noted (λ_em_ = 459–514 nm for **L1** and λ_em_ = 453–514 nm for **L2**). (Figure 8 and Appendix A). In the **L1** and **L2**, hypsochromic shifts of the emission bands, together with the increase in the solvent polarity, were recorded. It can be related to the geometry distortion in the excited state, which implies a decrease in the resonance energy. This feature is important, because the tailored emission can lead to potential applications as optical sensors. Similar behavior was noted for the ligands obtained also from o-phenylenediamine [30].

The emission bands in **L2** spectrum registered after excitation at 350 nm were split into two components in benzene.

The quantum yields of studied ligands are high or very high: 0.70 for **L1** in MeOH and 0.59 **L2** in acetonitrile. In general, the quantum efficiency is between 0.14 and 0.70. The lowest values of φ in MeOH for **L1** and benzene for **L2** were registered. The increase in the quantum efficiency with the solvent polarity in the case of **L1** was noted (λ_ex_ = 350 nm). For **L2,** this trend is similar, except for the chloroform. The opposite trend was observed by us in the series of complexes obtained from o-phenylenediamine and a series of various aldehydes. This tendency was connected with the loss of planarity in the excited state in polar solvent owing to an increase in the non-radiative processes [40]. The emission spectra of the ligands in the solid state exhibited emission bands at 548 nm for **L1** and 549 nm for **L2** (λ_ex_ = 295 nm). Moreover, excitation at 350 nm led to the emission band at 552 nm for **L1** and 561 nm for **L2**, respectively. For both ligands, a bathochromic shift of the emission bands in comparison with the solution was registered. (Figure 9, Appendix A). Red shifting of emission maxima was observed for many fluorescent compounds in the solid state due to π–π stacking of the aromatic rings in the molecules [41]. When the emission spectra registered in a solution and in the solid state are compared, it is possible to infer that the solvent destroys the π–π interactions, so the transition energy is increased in the solution.

Finally, the compounds are luminescent, both in the solution and in the solid state, which could be of significance in the search for new LEDs and can be considered as candidates for optical devices.

### 2.4. Thin Materials of Macrocyclic Ligands

The thin materials were obtained in two ways: by spin coating and by thermal evaporation. The morphology and the surface roughness of the thin films were investigated by SEM and AFM techniques. In order to study the chemical composition of the films, EDS analyses were conducted for all the samples.

#### 2.4.1. Thin Layers Obtained by Spin Coating

The optimum parameters of the layers (roughness, thickness, and homogeneity) obtained in the multistage spin coating process, were as follows: the spin speed 3000 rpm, time of coating 5 s for **L1**, (Figure 10) and 2500 rpm and 10 s for **L2**. (Figure 11) The two-dimensional (2D) and three-dimensional (3D) AFM images scanned over a surface area of 1 × 1 µm^2^ are shown in Figure 10 and Figure 11. The root-mean-square (RMS) parameters were calculated from the AFM images. The AFM images of the films indicate thin, amorphous layers of compounds deposited on the silicon surfaces with roughness parameters (of the deposited film) in the range R_a_ = 0.62–5.14 nm and R_q_ = 0.99–6.50 nm for **L1**/Si, and R_a_ = 0.39–3.76 nm and R_q_ = 1.70–6.36 nm for **L2**/Si. Occasionally, small crystallites appeared in the layer of **L2**/Si.

The EDS mapping confirmed the presence of the carbon, hydrogen, and oxygen on the entire silicon surface. (Figure 12—**L1**/Si and Figure 13—**L2**/Si).

#### 2.4.2. Thin Layers Obtained by Thermal Vapor Method

Materials obtained by thermal evaporation were thicker than those acquired by the spin coating technique. (Figure 14, Figure 15 and Figure 16) Results of SEM/EDS and AFM analysis reveal the presence of regular, thin, homogenous Schiff base materials, with height in the range of 5.8 and 6.5 nm. Moreover, SEM/EDS analysis showed the presence of carbon, nitrogen, and oxygen in the layer. (Figure 16 and Appendix A) SEM/EDS, together with mapping analysis, confirmed the composition of the material.

The new films were also characterized by IR DRIFT (Appendix A). The analysis of the IR DRIFT data showed the presence of the characteristic for the Schiff bases peaks between 1600 and 1594 cm^−1^ from stretching frequencies of the azomethine group, and bands from ν_Ph-O_ in the range of 1300–1275 cm^−1^. Moreover, the bands from stretching vibrations of aromatic rings ν_C=CAr_ in the region 1525–1425 cm^−1^ were registered. The above-described bands confirmed the presence of the deposited compounds in the obtained materials.

#### 2.4.3. Spectroscopic Ellipsometry Results of the Thin Materials

The optical constants and thicknesses of the deposited films were determined based on spectroscopic ellipsometry measurements using a four medium, optical model of a sample (from bottom to top): Si/native SiO_2_/layer (**L1** or **L2**) / ambient. The refractive index (*nI*) is a complex quantity *nI = n**-ik*. Quantities *n* and *k* are the real part of *nI* and the extinction coefficient, respectively. Optical constants of silicon and silicon dioxide were taken from the database of optical constants [42].

The complex refractive index of the layer was parameterized using the sum of Gaussian oscillators [42,43]:(1)$${ñ}^{2}\;=\varepsilon_{\infty }+\sum_{j}\mathrm{Gauss}(\mathrm{Aj},\;\mathrm{Bj},\;\mathrm{Brj})$$

In Equation (1) ε_∞_ is a high-frequency dielectric constant, while A_k_, E_k_, and Ba_rk_ are the amplitude, energy and broadening of an oscillator. The model parameters were adjusted to minimise the mean squared error (χ2), which is defined as [42,43]:
(2)$$\chi^{2}=\frac{1}{N-P}\sum_{j}\left({\left(\Psi_{j}^{mod}-\Psi_{j}^{exp\;}\right)}^{2}\;+\;{\left(\Delta_{j}^{mod\;}-\;{\Psi \Delta }_{j}^{exp\;}\right)}^{2}\;\right)$$
where *N* and *P* are the total number of data points and the number of fitted model parameters,
$$\Psi_{j}^{mod},\;\Psi_{j}^{exp},\Delta_{j}^{mod}\;and\;\Delta_{j}^{exp}$$
respectively, while the quantities are experimental (the quantities with superscript “exp”) and calculated (the quantities with superscript “mod”) ellipsometric azimuths.

An example of measured Ψ and Δ ellipsometric azimuths (for the **L1** sample) with model fits is presented in Figure 17a. The χ^2^ value was calculated to be 2.65. The determined thickness of the **L1** layer is 437 ± 2 nm, while for the **L2** film 772 ± 6 nm.

The optical constants (*n* and *k*) determined for the **L1** and **L2** films exhibit semiconducting behaviour (see Figure 17b,c). We did not observe such kind of feature in the materials obtained previously by us. For longer wavelengths (IR and partially vis), the extinction coefficient is equal to 0. The abundant absorption features are visible in the UV spectral range (see Figure 17c,d). In the non-absorbing spectral range, the refractive index shows normal dispersion relation. The band-gap energy (*E_g_*) was determined using the Tauc plot [44]. According to the following relation:(3)$$(\alpha h\nu )=B{\left(h\nu -E_{g}\right)}^{m}$$
to obtain the value of *E_g,_* the quantity ${\left(\alpha h\nu \right)}^{\frac{1}{m}}$ should be plotted as a function of *hν*. The exponent *m* depends on the type of transition and equals *m* = 1/2 for direct allowed transition, *m* = 3/2 for direct forbidden transition, *m* = 2 for indirect allowed transition and *m* = 3 for indirect forbidden transition [44]. The values of the absorption coefficient at the level of 10^5^ cm^−1^ suggest the direct transition [45] (m = 1/2). The Tauc plot for the obtained films is presented in Figure 17d. The band-gap energy is 3.45 ± 0.02 eV (359 ± 2 nm) and 3.29 ± 0.02 eV (377 ± 2 nm) for **L1** and **L2** samples, respectively. It should be noted that the absorption coefficient (and the extinction coefficient) for energies smaller than the value of *E_g_* exhibits relatively low or non-zero values (intra-ligand transitions). Considering this fact, the value of the band-gap energy should be treated as a value above, which a significant increase in the absorption coefficient takes place. The absorption features at about 340–350 nm (see Figure 17c,d) are bands related to the π→π* transitions associated with the azomethine group. The bands for wavelengths below 300 nm are π→π* transitions from the aromatic rings.

#### 2.4.4. Fluorescence of Thin Materials

The fluorescence properties of the **L1**/Si and **L2**/Si films were also studied. The observed emission at 560 nm (λ_ex_ = 350 nm) was connected to the IL transitions (Figure 18) and π→π* stacking interactions. The bathochromic shift of the fluorescence bands of the films in comparison to the solution was registered. Therefore, an influence of molecular packing in the solid phase on the optical properties can be concluded. The same was noted for the previous series of other compounds synthesized by us. The ligand layers **L1**/Si and **L2**/Si exhibited high fluorescence intensity.

Furthermore, the materials of the **L1**/Si and **L2**/Si exhibited broad emission. This feature can allow for using these compounds in LED diodes.

## 3. Materials and Methods

2-hydroxy-5-methylisophthalaldehyde (97%), o-phenylenediamine (analytical grade), 2-hydroxy-5-tert-butyl-1,3-benzenedicarboxaldehyde (97%) were purchased from Aldrich, Poland and used without further purification.

### 3.1. Methods and Instrumentation

^1^H, ^13^C, ^1^H ^13^C hmqc and hmbc NMR spectra of the Schiff bases were collected with a Bruker Avance III 400 MHz, Fallanden, Switzerland (**L2**) or a Bruker Avance 500 MHz spectrometer MA, USA (**L1**) in CDCl_3_. UV-vis absorption spectra were recorded on a Hitachi spectrophotometer, Fallanden, Switzerland in chloroform, acetonitrile, benzene (1.7 × 10^−5^ mol/dm^3^) and methanol (2 × 10^−5^ mol/dm^3^ and 1.4 × 10^−5^ mol/dm^3^) (grating 1, bandpass 8, integration time 100 ms). The fluorescence spectra were recorded on a spectrofluorometer Gildenpλotonics 700, Dublin, Ireland, in the range 700–200 nm (MeCN, chloroform, methanol and benzene solutions of compounds the same as in the case of UV-vis studies or silicon slides). The IR spectra were performed on the Bruker, MA, USA, using the ATR technique in the range of 400–4000 cm^−1^.

### 3.2. Crystal Structure Determination

The diffraction data of the studied compounds were collected for the single crystal at room temperature using an Oxford Sapphire CCD diffractometer, Oxford, UK, MoKα radiation λ = 0.71073 Å for **L1** and on Oxford Diffraction SuperNova DualSource diffractometer with monochromated Cu Kα X-ray source Oxford, UK, (λ = 1.54184 Å) for **L2**. For **L1**, the twinned data were processed with two domains using CrysAlis Pro [46]. Subsequently, the refinement cycles for **L1** were performed using HKLF 5 flag to take into account both domains. CCDC 2046259 and 2047048 contain the supplementary crystallographic data for **L1** and **L2**, respectively.

### 3.3. Computational Details

The theoretical calculations have been performed for the crystal structures, for **L1** and **L2**. The absorption spectra for all the systems have been calculated in a vacuum and in solvents (acetonitrile, methanol and chloroform) described with the polarizable continuum model in the linear response formalism. The B3LYP/6-311++G(d,p) approach has been applied for the geometry optimization, while for the vertical absorption calculations, the PBE0 functional has been applied. In the main text of the manuscript, only the results obtained in acetonitrile are given. The remaining results, together with corresponding numerical values of absorption wavelengths and oscillator strengths, are available in Appendix A. All the calculations were performed with Gaussian16 program (Frisch, M.J.; Trucks, G.W.; Schlegel, H.B.; Scuseria, G.E.; Robb, M.A.; Cheeseman, J.R.; Scalmani, G.; Barone, V.; Petersson, G.A.; Nakatsuji, H.; et al Revision B.01; Gaussian, Inc.: Wallingford, UK, 2016). [47].

### 3.4. Thin Materials

#### 3.4.1. Spin Coating

Layers of the compounds **L1** and **L2** were deposited on Si(111) wafers (10 nm × 10 mm) ~500 nm thick using the spin coating technique. Precursors were dissolved in DMSO and deposited on Si using a spin coater (Laurell 650 SZ, US). The spin speed varied from 2500 rpm to 3000 rpm, the coating time was 5 or 10 s.

#### 3.4.2. Thermal Vapor Deposition

The thin layer of **L1** and **L2** was deposited on n-type silicon substrate. The orientation of the silicon substrate was (100) with electrical resistivity (ρ) equal to 6.2 × 10^−3^ Ω cm. The silicon wafer was first degreased in acetone, ethanol, and finally in deionized water using an ultrasonic bath. On the front side (polished side) of the silicon wafer, a **L1** and **L2** layer of 4–5 nm thickness were deposited in a vacuum (p = 2 × 10^−4^ Pa) by a thermal evaporation method, with an evaporation rate of 0.2 nm/s, without heating of the substrate.

### 3.5. Spectroscopic Ellipsometry

Ellipsometric measurements were performed using the V-VASE device (from J.A.Woollam Co., Inc.; Lincoln, NE, USA). Ellipsometric azimuths (Ψ and Δ) were recorded in the spectral range from 0.6 eV (2000 nm) to 6.0 eV (225 nm) for three angles of incidence (65 °C, 70 °C and 75 °C).

### 3.6. Morphology and Composition of the Materials

The morphology and composition of the obtained films were analyzed with a scanning electron microscope (SEM), LEO Electron Microscopy Ltd., England, the 21430 VP model equipped with secondary electrons (SE) detectors and an energy dispersive X-ray spectrometer (EDX) Quantax with an XFlash 4010 detector (Bruker AXS microanalysis GmbH, England). The layers morphology was also studied using SEM/FIB (scanning electron microscope/focused ion beam) Quanta 3D FEG equipped with gold and palladium splutter SC7620, Czechoslovakia. The atomic force microscopy (AFM) images were taken in the tapping mode with a Multi Mode Nano Scope IIIa (Veeco Digital Instrument, SB, USA) microscope.

### 3.7. Synthesis of Compounds

#### 3.7.1. **L1**

0.1636 g (0.001 mol) of 2-hydroxy-5-methyl-1,3-benzenedicarboxaldehyde was added to 0.1083 g (0.001 mol) of o-phenylenediamine dissolved in 50 cm^3^ of methanol. The synthesis was carried out under reflux for 2 h; the product was filtered off. The product was dried under air; orange single crystals were received, and the yield of the synthesis was 50%. The melting point of the obtained product was 270–273 °C. C_30_H_28_N_4_O_2_ (calc./found %): C 75.61/75.64, N 11.76/11.52, H 5.92/5.95.

^1^H [ppm]: 2.30 (s, 6H) (H1), CH_3_, 4.40 (s, 4H) (H6), CH_2_, 6.31 (s, 2H) Ar-H (3), 6.75–6.78 (td, 2H, *J* = 1.16 Hz, *J* = 7.5 Hz) Ar-H (9), 6.92–6.94 (dd, 2H, *J*= 8.2 Hz, 1 Hz) Ar-H (10), 7.01–7.03 (dd, 2H, *J* = 8 Hz, 1.5 Hz) Ar-H (8), 7.13–7.14 (d, 2H, *J* = 1,5 Hz) Ar-H (15), 7.17–7.18 (d, 2H, *J* = 1.9 Hz) Ar-H (11), 8.55 (s, 2H) -N=CH-, 13.5 (s, 2H) -OH, (Appendix A) ^13^C [ppm]: 20.36 (C1), 46.91 (C6), 111.78 (C8), 117.44 (C14), 117.99 (C9), 119.55 (C10), 125.33 (C4), 128.18 (C11), 128.30 (C3), 131.57 (C15), 134.28 (C7), 136.11 (C12) 143.49 (C2) 157.36 (C5) 161.76 (C13) [33,34] (Appendix A). Selected FT-IR (data reflectance, crystal) (cm^−1^) 3405 ν_OH_, 3070, 2634 ν_C-HAr_, 1600, 1587 ν_C=N_, 1508 ν_C=CHA_, 1467, 1367 ν_C=CAr_, 1323 ν_C-NHA_, 1288 ν_Ph-O_. (Appendix A) UV-vis (MeCN, 1.7 × 10^−5^ mol/dm^3^): 338 (16,176), 346 (16,294), 382 (16,353) (chloroform, 1.7 × 10^−5^ mol/dm^3^): *λ*/nm 338 (*ε*/dm^3^ mol^−1^ cm^−1^ 16,765), 350 (17,117), 382 (15,823), (MeOH, 2 × 10^−5^ mol/dm^3^): *λ*/nm 342 (*ε*/dm^3^ mol^−1^ cm^−1^ 16,800), (benzene, 1.7 × 10^−5^ mol/dm^3^): *λ*/nm 336 (*ε*/dm^3^ mol^−1^ cm^−1^ 14,529), 350 (14,765), 390 (14,412).

#### 3.7.2. **L2**

0.1080 g (0.001 mol) of o-phenylenediamine was added to 0.2055 g (0.001 mol) of 2-hydroxy-5-tert-butyl-1,3-benzenedicarboxaldehyde dissolved in 50 cm^3^ of methanol. The synthesis was carried out under reflux for 2 h, the product was filtered off. The product was dried under the air. Brown single crystals were received. (Yield: 60%). m.p.: 275–280 °C. C_36_H_40_N_4_O_2_ (calc./found %): C 77.11/77.26, N 9.99/10.31, H 7.19/7.29.

^1^H [ppm]: 1.35 (s, 18H) (H1), CH_3_, 4.48 (s, 4H) (H7), CH_2_, 6.81–6.82 (t, 2H, *J* = 7.2 Hz) Ar-H (4), 7.00–7.01 (d, 2H, *J* = 7.7 Hz) Ar-H (11), 7.08–7.09 (dd, 2H, *J* = 7.6 Hz, 1.3 Hz)) Ar-H (10), 7.27–7.28 (dd, 2H, *J* = 15.5 Hz, 1.6 Hz) Ar-H (9), 7.37 (d, 2H, *J* = 2.4 Hz) Ar-H (12), 7.45 (d, 2H, *J* = 2.4 Hz) Ar-H (16), 8.65 (s, 2H) -N=CH-, 13.57 (s, 2H) -OH, (Appendix A) ^13^C [ppm]: 31.42 (C1), 34.05 (C2), 47.37 (C7), 111.90 (C9), 117.56 (C15), 117.98 (C10), 119.21 (C11), 124.99 (C16), 128.05 (C4, C6, C5), 128.24 (C12), 130.88 (C8), 136.24 (C13), 141.89 (C3), 157.26 (C6), 162.15 (C14) (Appendix A) [33,34].

Selected FT-IR (data reflectance, crystal) (cm^−1^) 3423 ν_OH_, 2958, 2861 ν_C-HAr_, 1600, 1587 ν_C=N_, 1508 ν_C=CHA_, 1455, 1364 ν_C=CAr_, 1333 ν_C-NHA_, 1289 ν_Ph-O_. (Appendix A) UV-vis (MeCN, 1.7 × 10^−5^ mol/dm^3^): *λ*/nm 340 (*ε*/dm^3^ mol^−1^ cm^−1^ 19,529), 382 (20,882), (chloroform, 1.7 × 10^−5^ mol/dm^3^): *λ*/nm 338 (*ε*/dm^3^ mol^−1^ cm^−1^ 27,882), 346 (28,471), 380 (27,176), (MeOH, 1.4 × 10^−5^ mol/dm^3^): *λ*/nm 342 (*ε*/dm^3^ mol^−1^ cm^−1^ 22,214), (benzene, 1.7 × 10^−5^ mol/dm^3^): *λ*/nm 342 (*ε*/dm^3^ mol^−1^ cm^−1^ 16,941), 390 (16,176).

## 4. Conclusions

Two macrocyclic Schiff bases, **L****1** and **L****2**, derived from o-phenylenediamine and 2-hydroxy-5-methylisophthalaldehyde **L1** or 2-hydroxy-5-tert-butyl-1,3-benzenedicarboxaldehyde **L2**, respectively, were obtained. The X-ray structures for the compounds were isolated.

The studies of the fluorescence properties of the compounds showed that the excitation of **L1** at 295 nm leads to the emission in the range of 454–516 nm, and of **L2** to the emission between 452 and 517 nm in dependence on the solvent polarity. In the **L1** and **L2**, hypsochromic shifts of the emission bands, together with the increase in the solvent polarity, were recorded.

For both compounds, the bathochromic shift of the emission bands in the solid state compared to the solution was registered. The materials obtained by thermal vapor exhibited fluorescence at 560 nm (λ_ex_ = 350 nm), which was connected to the IL transitions.

The high intensity of the fluorescence indicated **L1**/Si and **L2**/Si materials, which can be used in the optical devices. The ellipsometric analysis of the new materials obtained by the thermal vapor technique showed that the band-gap energy was 3.45 ± 0.02 eV (359 ± 2 nm) and 3.29 ± 0.02 eV (377 ± 2 nm) for **L1** and **L2** samples, respectively. Moreover, the **L1** and **L2** films exhibited semiconducting behaviour. The values of the roughness parameters indicate the achievement of smooth films of macrocyclic Schiff bases **L1**/Si and **L2**/Si. This is important because the layers which can be used, e.g., in OLEDs have to be smooth and thin. Our materials can be also used as semiconductors.

The computational procedure used allowed for the prediction of the relative tendencies in the absorption spectra for all the compounds and the determination of the character of the transitions in the spectra of all the isolated compounds.

## Figures and Tables

**Figure 1 molecules-27-07396-f001:**
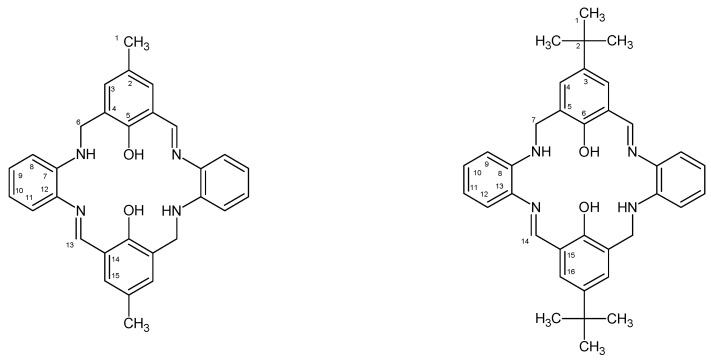
Structural formulas of **L1** (**left**) and **L2** (**right**) ligands with the numbering scheme.

**Figure 2 molecules-27-07396-f002:**
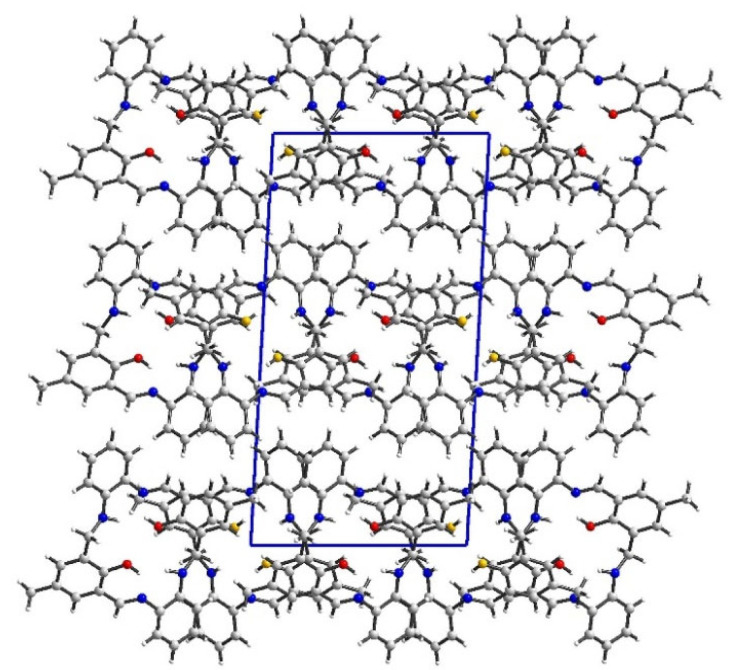
Packing of **L1** along b axis. O14 atoms are marked in red and O34 in orange to indicate motifs observed in the crystal network.

**Figure 3 molecules-27-07396-f003:**
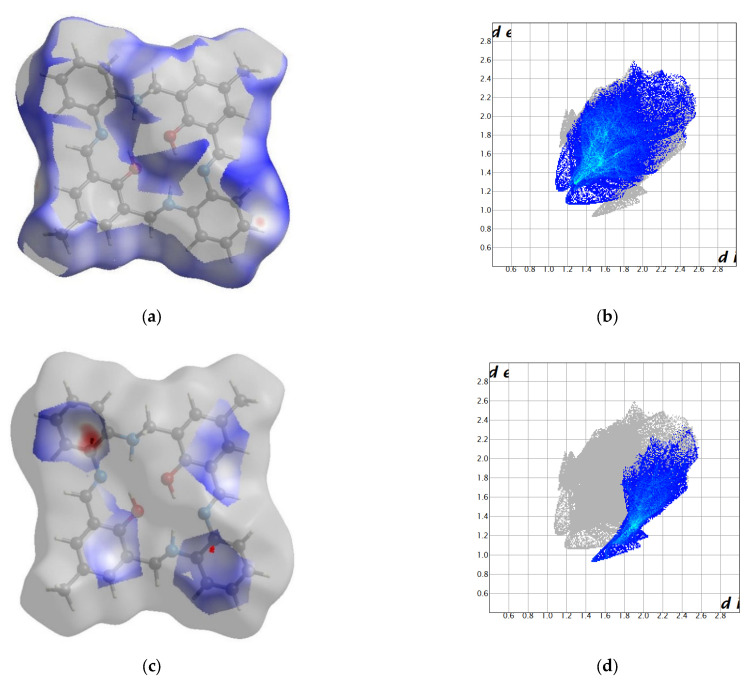
Hirshfeld surfaces and fingerprints of selected interactions created in the crystal network of **L1**: (**a**) Hirshfeld surface for H…H; (**b**) fingerprint for H…H (50.7%); (**c**) Hirshfeld surface for C…H (red markers correspond to the large spike at 1.5; 0.9); (**d**) fingerprint for C…H (19.9%); (**e**) Hirshfeld surface for H…C; (**f**) fingerprint for H…C (15.1%) for O14 molecule In brackets, there is given surface area included as a percentage of the total surface area.

**Figure 4 molecules-27-07396-f004:**
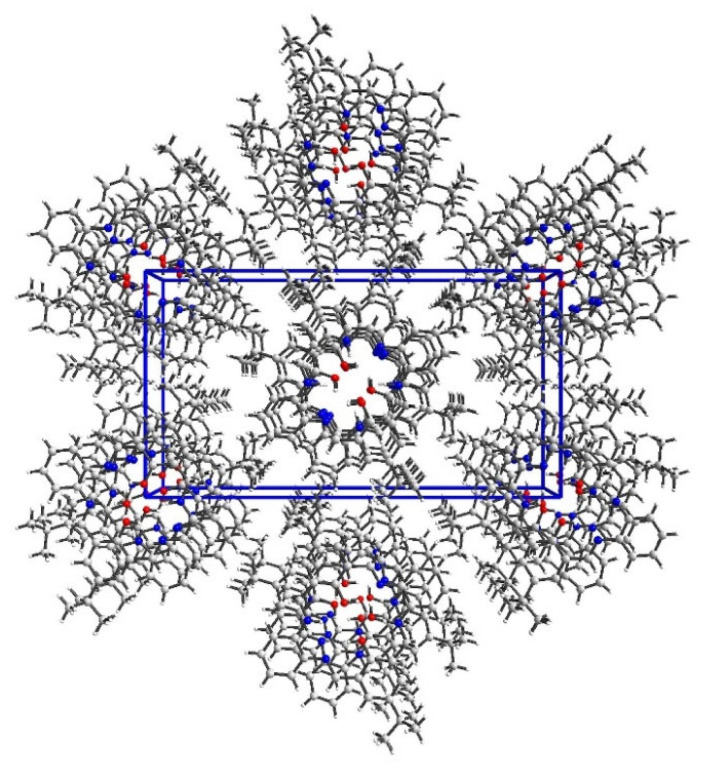
Perspective view of **L2** along c axis shows a column composed of alternately arranged O14 and O34 molecules rotated by ca. 46°.

**Figure 5 molecules-27-07396-f005:**
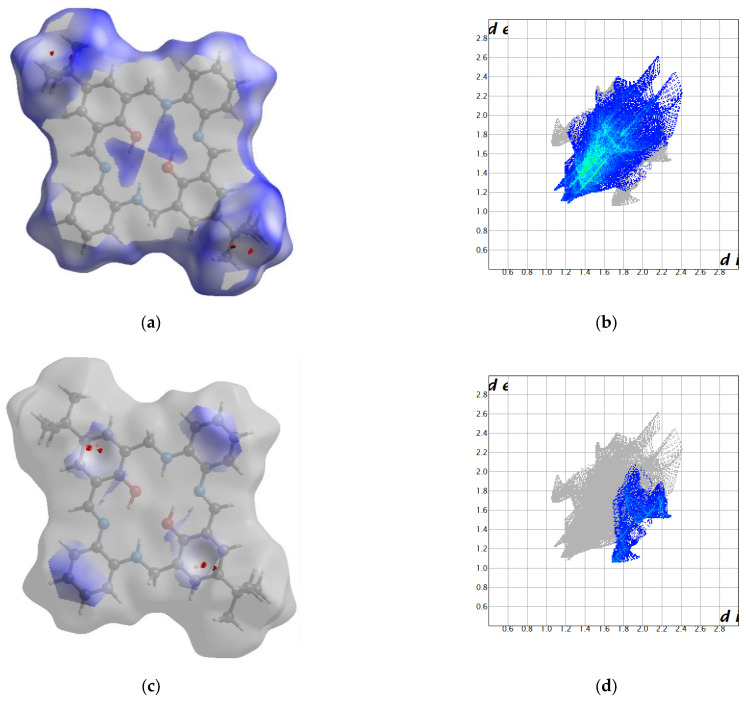
Hirshfeld surfaces and fingerprints of selected interactions created in the crystal network of **L2**: (**a**) Hirshfeld surface for H…H (red markers correspond to the spike at 1.15, 1.1); (**b**) fingerprint for H…H (65.4%); (**c**) Hirshfeld surface for C…H (red markers correspond to the spike at 1.7, 1.1); (**d**) fingerprint for C…H (12.4%); (**e**) Hirshfeld surface for H…C (red markers correspond to the spike at 1.1, 1.7); (**f**) fingerprint for H…C (10.6%) for O14 molecule In brackets, there is given surface area included as a percentage of the total surface area.

**Figure 6 molecules-27-07396-f006:**
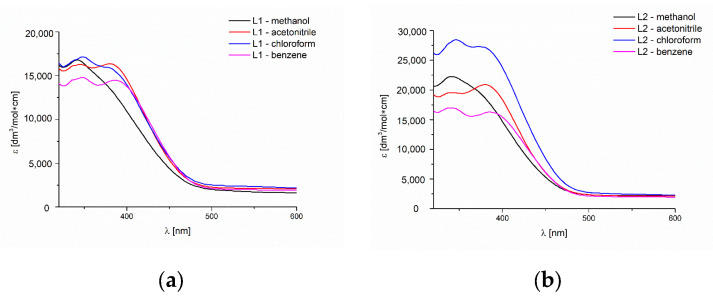
Solution absorption spectra of (**a**) **L1** (**b**) **L2** in chloroform, acetonitrile, benzene (1.7 × 10^−5^ mol/dm^3^, RT) and methanol (2 × 10^−5^ mol/dm^3^, RT) for **L1**, (1,4 × 10^−5^ mol/dm^3^, RT) for **L2**.

**Figure 7 molecules-27-07396-f007:**
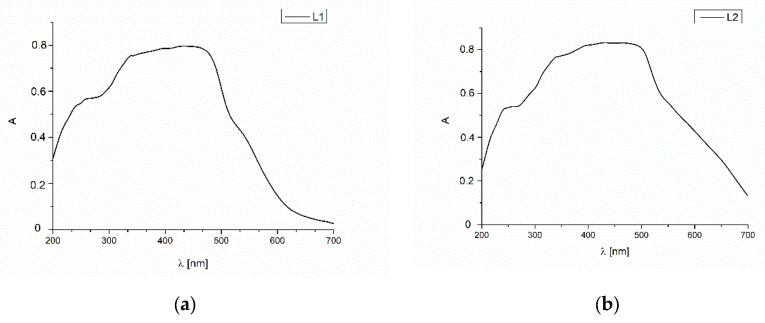
Solid state absorption spectra od (**a**) **L1** and (**b**) **L2**.

**Figure 8 molecules-27-07396-f008:**
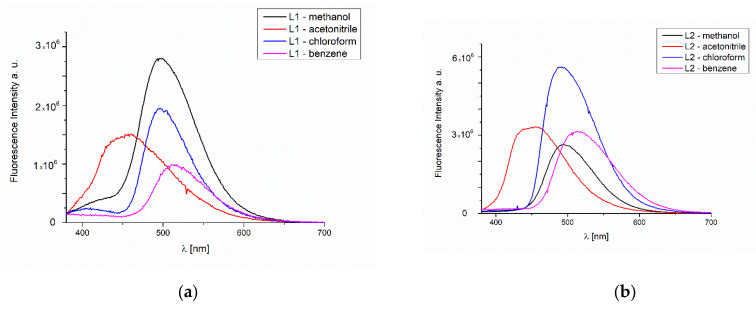
Solution emission spectra of **L1** and **L2**. (**a**) **L1** (**b**) **L2** *λ*_ex_ = 295 nm, (**c**) **L1** (**d**) **L2** *λ*_ex_ = 350 nm, (chloroform, acetonitrile, benzene (1.7 × 10^−5^ mol/dm^3^, RT) and methanol (2 × 10^−5^ mol/dm^3^, RT) for **L1**, (1,4 × 10^−5^ mol/dm^3^, RT) for **L2**.

**Figure 9 molecules-27-07396-f009:**
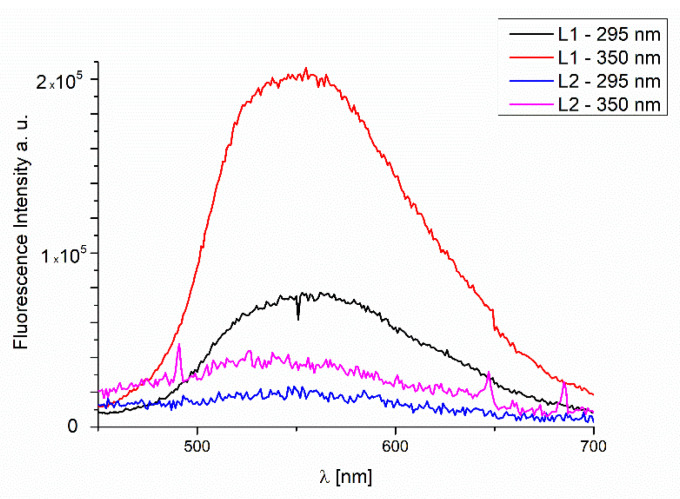
Solid state emission spectra of ligands **L1** and **L2**; *λ*_ex_ = 295 nm, *λ*_ex_ = 350 nm.

**Figure 10 molecules-27-07396-f010:**
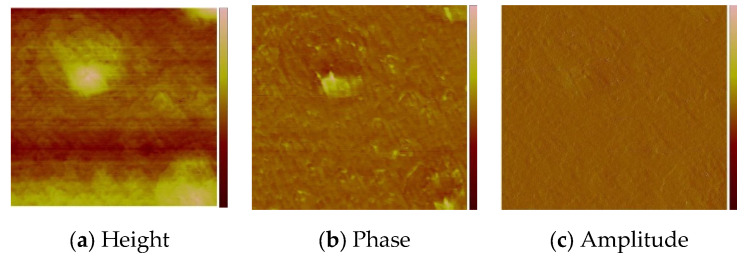
AFM of **L1**/Si; 3000 rpm, 5s × 10, scan size 1 μm, R_a_ = 0.62 nm, R_q_ = 0.99 nm.

**Figure 11 molecules-27-07396-f011:**
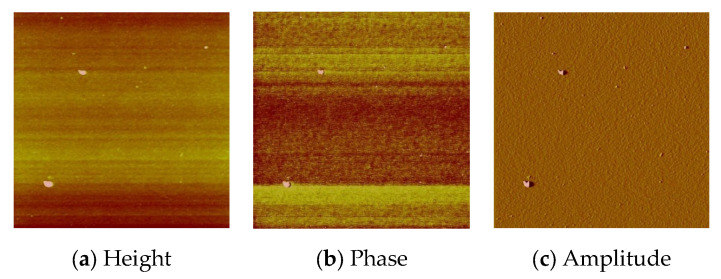
AFM of **L2**/Si; 2500 rpm, 10s × 15, scan size 10 μm, R_a_ = 1.75 nm, R_q_ = 2.22 nm.

**Figure 12 molecules-27-07396-f012:**
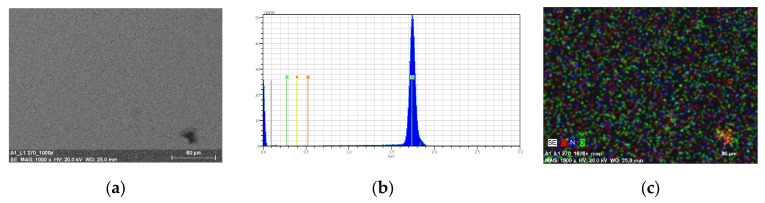
SEM of **L1**/Si (**a**) **L1**/Si; 3000 rpm 5s × 10 (**b**) EDS and (**c**) mapping of **L1**/Si, 3000 rpm 5s ×10, scan size 1 µm.

**Figure 13 molecules-27-07396-f013:**
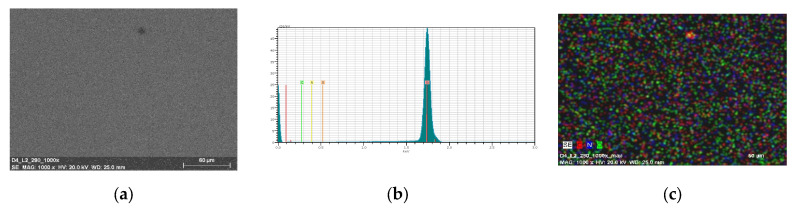
SEM of **L2**/Si spin coating (**a**) **L2**/Si, 2500 rpm 10s ×15 (**b**) EDS and (**c**) mapping of **L2**/Si, 2500 rpm 10s × 15, scan size 1 µm.

**Figure 14 molecules-27-07396-f014:**
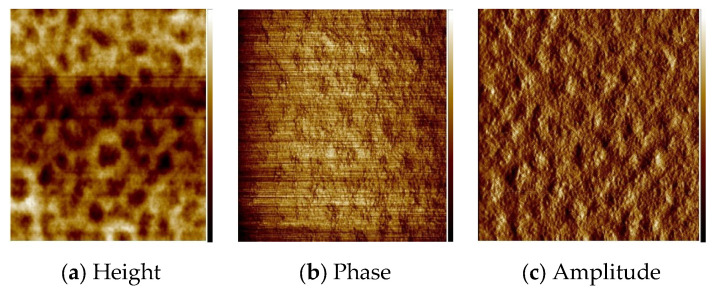
AFM of **L1**/Si, R_a_ = 1.32 nm, R_q_ = 1.78 nm, thermal vapor deposition, scan size 5 µm.

**Figure 15 molecules-27-07396-f015:**
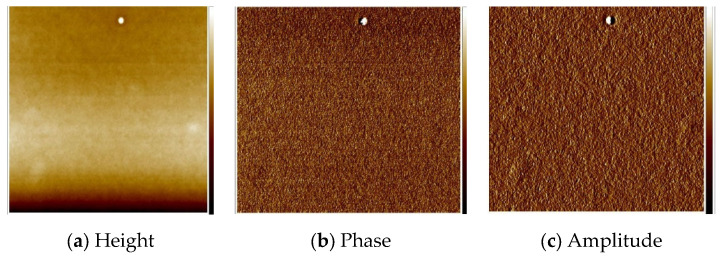
AFM of **L2/**Si, R_q_ = 4.87 nm, R_a_ = 3.63 nm, thermal vapor deposition, scan size 10 µm.

**Figure 16 molecules-27-07396-f016:**
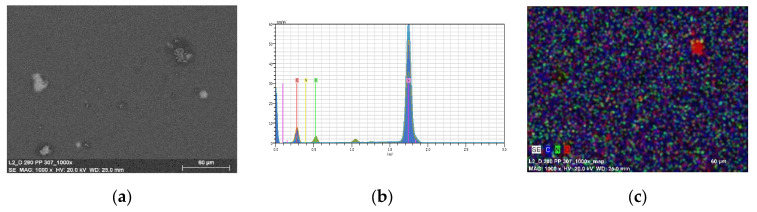
(**a**) SEM of **L2/**Si from thermal vapor deposition; (**b**) EDS and (**c**) mapping.

**Figure 17 molecules-27-07396-f017:**
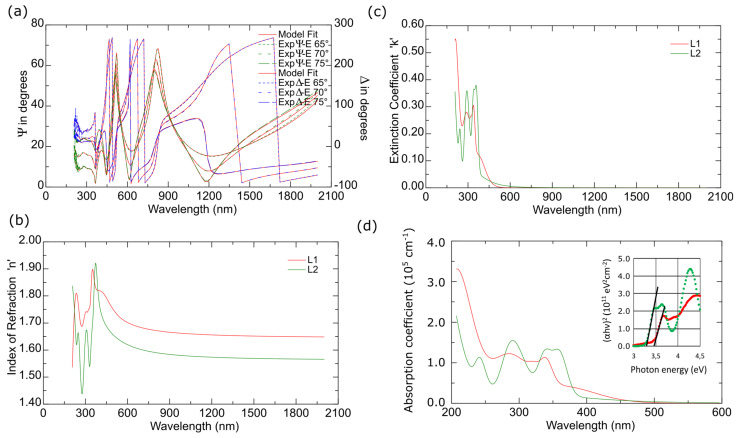
(**a**) Experimental and calculated Ψ and Δ azimuths for the **L1** sample; (**b**) The refractive index (*n*); (**c**) the extinction coefficient (*k*); (**d**) the absorption coefficient of evaporated **L1** and **L2** layers (inset: the Tauc plot).

**Figure 18 molecules-27-07396-f018:**
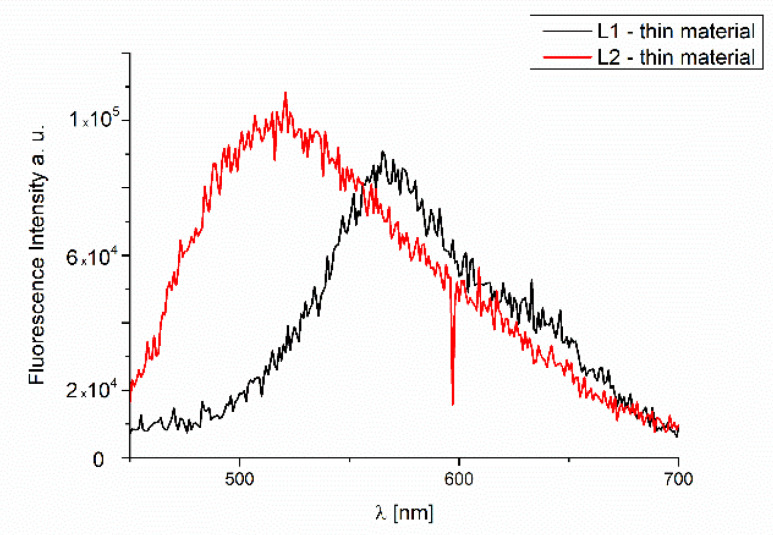
Fluorescence of **L1** and **L2** materials obtained via thermal vapor deposition.

## Data Availability

The data presented in this study are available on request from the corresponding author.

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
