# Peer review of "Experimental and Theoretical Studies of the Optical Properties of the Schiff Bases and Their Materials Obtained from o-Phenylenediamine"

_molecules, 2022, doi:10.3390/molecules27217396_

Round 1
Reviewer 1 Report
The author reported the synthesis of two macrocyclic Schiff ligands through the reaction of o-phenylenediamine and 2-hydroxy-5-methylisophthalaldehyde L1 or 75 2-hydroxy-5-tertbutyl-1,3-benzenedicarboxaldehyde. The experimental and theoretical studies of the compounds are good. The paper can be accepted for publication after minor modification.
1. Some of figures should be presented in Supporting Information.
Part 3 and Part 4 should be integrated in one part or removed to the Supporting Information.
3. The possible application of the two compounds should be discussed in Introduction or Conclusion.
4. The performances of the compounds for possible application should be compared with those of the previously reported compounds.
Author Response
I would like to thank the Reviewers for their remarks concerning problems found in my paper.
Answers for Reviewers comments
We are very grateful to the Reviewer for his/her valuable comments which enabled us to improve our model in terms of chemical credibility. Below we present the Reviewer comments addressed point by point. We strongly believe that the introduced changes improving our model and explanations should complete Reviewer requirements. In case of any doubts, please do not hesitate to contact us. All changes were marked in purple.
REVIEWER REPORT(S):
Referee: 1
The author reported the synthesis of two macrocyclic Schiff ligands through the reaction of o-phenylenediamine and 2-hydroxy-5-methylisophthalaldehyde L1 or 75 2-hydroxy-5-tertbutyl-1,3-benzenedicarboxaldehyde. The experimental and theoretical studies of the compounds are good. The paper can be accepted for publication after minor modification.
Comment 1: Some of figures should be presented in Supporting Information.
Answer 1: Figure 3g- 3l (now S9) and 4g-4l (now S10) were moved to the Supporting Information.
Comment 2: Part 3 and Part 4 should be integrated in one part or removed to the Supporting Information.
Answer 2:
It was changed, and now all data are placed in Part 3 in accordance with the instructions available on the website: https://www.mdpi.com/journal/molecules/instructions
Research Manuscript Sections
Materials and Methods: They should be described with sufficient detail to allow others to replicate and build on published results. New methods and protocols should be described in detail while well-established methods can be briefly described and appropriately cited. Give the name and version of any software used and make clear whether computer code used is available. Include any pre-registration codes.
Comment 3: The possible application of the two compounds should be discussed in the Introduction or Conclusion.
Answer 3:
It was changed.
Comment 4: The performances of the compounds for possible application should be compared with those of the previously reported compounds.
Answer 4:
It was compared.
Reviewer 2 Report
The manuscript entitled "Experimental and theoretical studies of the optical properties of the Schiff bases and their materials obtained from o-phenylenediamine" details the synthesis of two macrocyclic Schiff bases. These have been fully characterised by single crystal X-ray diffraction and spectroscopic methods to confirm the identify of the material. The discussion of the crystal structure is slightly hard to follow. Both complexes appear to be position around symmetry elements in the crystal structure and so only half of the molecule s present in the asymmetric unit (for L1 there are two half molecules leading to the O14/O34 labelling). This was not clear on reading the text and then the same terminally is used for L2 but this only have one half molecule in the asymmetric cell so there is not a O14/O34 separation present. Have you analysed the structures for void space?, Figure 4 hints are possibility of channels but this may be illusionary by the view and using ball and stick models.
The remaining section cover the electronic spectroscopy of the system and preparation of thin film samples and their characterisation. The experimental and computational work are well done. I do find figures 12,13 and 16 a bit small to read clearly, does the EDS mapping scale in part (b) have to run to 20 when the last peaks are at 2?
It is proposed that the system forms an amorphous phase in the thin film case, would the properties of this phase be different from the crystal forms? Can bulk amorphous phase be created to generate the spectra?
There are some typographic errors in the text and the introduction has focus on metal complexes of these ligands rather than the metal free system studied here. Are there any previous studies on similar systems that could be referenced?
Author Response
I would like to thank the Reviewers for their remarks concerning problems found in my paper.
Answers for Reviewers comments
We are very grateful to the Reviewer for his/her valuable comments which enabled us to improve our model in terms of chemical credibility. Below we present the Reviewer comments addressed point by point. We strongly believe that the introduced changes improving our model and explanations should complete Reviewer requirements. In case of any doubts, please do not hesitate to contact us. All changes were marked in purple.
REVIEWER REPORT(S):
The manuscript entitled "Experimental and theoretical studies of the optical properties of the Schiff bases and their materials obtained from o-phenylenediamine" details the synthesis of two macrocyclic Schiff bases. These have been fully characterised by single crystal X-ray diffraction and spectroscopic methods to confirm the identify of the material.
Comment 1:
The discussion of the crystal structure is slightly hard to follow. Both complexes appear to be position around symmetry elements in the crystal structure and so only half of the molecule s present in the asymmetric unit (for L1 there are two half molecules leading to the O14/O34 labelling). This was not clear on reading the text and then the same terminally is used for L2 but this only have one half molecule in the asymmetric cell so there is not a O14/O34 separation present. Have you analysed the structures for void space?, Figure 4 hints are possibility of channels but this may be illusionary by the view and using ball and stick models.
Answer 1: In the discussion the molecule description is missing because these structures are known and we focused on network interactions which were not provided in the original papers applying also Hirshfeld approach. However, we deposited those structures (CCDC 2046259 and 2047048 for L1 and L2, respectively) because we used them for theoretical calculations.
You are right, in L1 crystallizing in the monoclinic P21/c space group there are two halves of the molecule in the asymmetric unit positioned around the inversion center, Hence, in the crystal network two molecules differ slightly in conformation and in the created intermolecular interactions. Similarly, in L2 crystallizing in the orthorhombic Pbam space group there are two halves of the molecule in the asymmetric unit positioned around the inversion center. However, in this case all atoms of the macrocyclic ring are positioned also in the m^Z plane. It results in ideal planarity of this macrocycle due to symmetry reasons.
Both structures are tightly packed and any voids are not detected by PLATON. It is also marked by packing index.
Comment 2: The remaining section cover the electronic spectroscopy of the system and preparation of thin film samples and their characterisation. The experimental and computational work are well done.
I do find figures 12,13 and 16 a bit small to read clearly, does the EDS mapping scale in part (b) have to run to 20 when the last peaks are at 2?
Answer 2
It was changed. The figures are now bigger and scale was modified.
Comment 3: It is proposed that the system forms an amorphous phase in the thin film case, would the properties of this phase be different from the crystal forms? Can bulk amorphous phase be created to generate the spectra?
Answer 3:
The thin films were analysed by SEM, EDX, AFM, DRIFT and ellipsometry. The EDX as qualitative analysis confirmed the presence of N,C, O atoms in the layers.
- a) The SEM images showed the amorphous layers; in some parts the small crystallites as impurities can be noted. (Figure 16), what does not influence the properties of the amorphous films like it was noted before (e.g. Barwiolek, D. Jankowska, M. Chorobinski, A. Kaczmarek-Kedziera, I. Łakomska, S. Wojtulewski, T. M. Muzioł, RSC Adv. 2021, 11, 24515).
- b) Moreover, the IR DRIFT data showed the presence of the characteristic for the Schiff bases peaks between 1600-1594 cm-1 from stretching frequencies of the azomethine group, and bands from νPh-O in the range of 1300-1275 cm−1. Additionally, the bands from stretching vibrations of aromatic rings νC=CAr in the region 1525-1425 cm−1 were registered.
c)At the IR spectra of solid samples peaks at 1600 and 1587 cm−1 from stretching vibrations of the azomethine group. A sharp band at 3405 cm−1 for L1 and 3423 cm-1 for L2, can be ascribed to the vNH groups. The peaks at 3405, 3423 cm−1 also are attributed to the OH group vibrations. Additionally, bands from Ph-O stretching vibrations in the range of 1289-1288 cm−1 were noted. (Figure S14)
The similarity of both spectra confirmed the identity of the deposited compounds in the obtained materials with synthtised compounds.
- d) Ellipsometry analysis of the materials also indicates their amorphous nature. Moreover, the presence of the specific bands in the spectra also confirms the existence of the thin layer of the Schiff base.
Comment 4: There are some typographic errors in the text and the introduction has focus on metal complexes of these ligands rather than the metal free system studied here. Are there any previous studies on similar systems that could be referenced?
Answer 4
It was changed. Introduction was modyfying. Some references have been added.
Reviewer 3 Report
The authors explained the work nicely but I observed that the introduction part must be improved with some more relevant work. So authors are advised to rewrite the introduction part in the light of suggested articles i.e. doi.org/10.1016/j.molstruc.2022.133361; doi.org/10.1016/j.molstruc.2022.133939 ;
Regarding crystal structures of the ligands cannot comment as I was not able to access the CIF file as no CCDC numbers are mentioned in the manuscript, and also not access the supplementary file.
Author Response
I would like to thank the Reviewers for their remarks concerning problems found in my paper.
Answers for Reviewers comments
We are very grateful to the Reviewer for his/her valuable comments which enabled us to improve our model in terms of chemical credibility. Below we present the Reviewer comments addressed point by point. We strongly believe that the introduced changes improving our model and explanations should complete Reviewer requirements. In case of any doubts, please do not hesitate to contact us. All changes were marked in purple.
Referee:
The authors explained the work nicely but I observed that the introduction part must be improved with some more relevant work.
Comment 1: So authors are advised to rewrite the introduction part in the light of suggested articles i.e. doi.org/10.1016/j.molstruc.2022.133361; doi.org/10.1016/j.molstruc.2022.133939 ;
Answer 1
It was changed.
Comment 2: Regarding crystal structures of the ligands cannot comment as I was not able to access the CIF file as no CCDC numbers are mentioned in the manuscript, and also not access the supplementary file.
Answer 2
CCDC 2046259 and 2047048 contain the supplementary crystallographic data for L1 and L2, respectively. They were not given in the manuscript because these structures have been already known. However, you are right – this information is missing and we add such sentence to the manuscript also because they were used for theoretical calculations.
Author Response
I would like to thank the Reviewers for their remarks concerning problems found in my paper.
Answers for Reviewers comments
We are very grateful to the Reviewer for his/her valuable comments which enabled us to improve our model in terms of chemical credibility. Below we present the Reviewer comments addressed point by point. We strongly believe that the introduced changes improving our model and explanations should complete Reviewer requirements. In case of any doubts, please do not hesitate to contact us. All changes were marked in purple.
Comment 1:
This paper reports the experimental and theoretical studies of the optical properties of two macrocycle L1 and L2 and their L1/Si and L2/Si materials. The characterization work appears competently done. I agree with the authors that L1/Si and L2/Si materials can be of interest due their fluorescence properties, with potential applications in the developments for optoelectronic materials. On the other hand, I cannot see sufficient elements of novelty in the presented studies. The synthetic approach to obtain macrocycle L1 and L2, which are differ only by methyl or t-butyl substituent, have been known for several years and their spectroscopic properties were reported.
Answer
- Yes it’s true. It is “only” small change from methyl or t-butyl substituent, but as we were able to show even such a small change can have a huge influence on the properties of the obtained compounds. e.g. The 3D Hirschfeld analyses show that the most numerous interactions were found between hydrogen atoms. A considerable number of such interactions is justified by the presence of bulk tert-butyl groups in L2.
Similarly, determination of crystal structures for both compounds L1 and L2 do not add elements of novelty as their crystal structures have been already deposited in CCDC. I assume that the reason for which the authors reported all of it one more time was to confirm the structure of synthesized compounds and determine their purity. Thus, I cannot suggest publishing this manuscript in Special Issue: “Design and Synthesis of Macrocyclic Compounds” I reckon that the detailed studies of the L1/Si and L2/Si morphology layers as well as the fluorescence properties of L1/Si and L2/Si materials are valuable but should be published in other journals, which aim and scope cover nanotechnology and properties of new materials e.g. Materials.
Answer
The novelty of our work is focused on:
- a) the optical properties (Uv-Vis and fluorescence) of the obtained ligands in the solution (different solvents) and in the solid state.
- b) the optical properties (Uv-Vis and fluorescence) and ellipsometry of the thin materials.
- c) DFT calculations for that kind of compounds.
What is also important in our work we focus are concentrate on the macrocyclic ligands and their properties, while most of the papers deals with the metal complexes with these compounds and their properties.
So, we fully believe that our publication fit to the Special Issue: “Design and Synthesis of Macrocyclic Compounds” It connects properties of the isolated macrocyclic compounds from solution towards the solid state to thin materials.
Comments:
Comment 2: There are several inconsistencies between the numbers of figures and their captions in the Supplementary Materials and the main text. Please carefully check the numbers of the supplemental figures and their captions.
Answer 2
It was changed.
Comment 3: By convention, 1H NMR data are listed from the lowest field values to the highest or the other way round, not from the right part of the spectrum, left and the middle. Moreover, in case of the doublet of doublets (dd), two values of coupling constants should be written. Line 443) there is 1,35 (s,9H); should be 18H Line 490) the word Figure is repeated: Figure Figure S10
Answer 3:
It was changed.
Comment 4: In References section:
Journal abbreviations should be written in italic (ref. numbers 3 and 11);
There is inconsistency in the journal citation (for the same journal in ref. 4 abbreviation is used
whereas in ref. 9 full title)
Line 518) revise the author names and surnames please
Line 599) There is inconsistency in the author names citation (Surname Name and Surname, N.)
Answer 4:
It was changed.
Round 2
Reviewer 4 Report
The authors confirmed that the novelty of their studies is the determination of the optical properties of L1/Si and L2/Si materials. The studies that the authors described do not deal with “Design and Synthesis of Macrocyclic Compounds”. The results are valuable but they represent only the outcome of the routine tests for optical properties determinations of new materials. In my opinion, the manuscript does not in line with the high standard of Molecules and can be published in any journal which covers the properties of new materials. The final decision rests with the Editor.
Author Response
Comment 1:
The authors confirmed that the novelty of their studies is the determination of the optical properties of L1/Si and L2/Si materials. The studies that the authors described do not deal with “Design and Synthesis of Macrocyclic Compounds”. The results are valuable but they represent only the outcome of the routine tests for optical properties determinations of new materials. In my opinion, the manuscript does not in line with the high standard of Molecules and can be published in any journal which covers the properties of new materials. The final decision rests with the Editor.
Answer
We synthesized and fully optically characterized not only thin materials but also macrocyclic ligands.
And, again what is also very important in our work is that we focus only on the macrocyclic ligands (Uv-Vis and fluorescence, DFT calculations) and their properties, while most of the papers deal with the metal complexes with these compounds and their properties. Additionally, the properties of the thin layers obtained by spin coating and thermal vapour deposition was studied.
So, we fully believe that our publication fit the Special Issue: “Design and Synthesis of Macrocyclic Compounds” It connects properties of the isolated macrocyclic compounds from solution towards the solid state to thin materials.